# The Effect of Relative Advantage, Top Management Support and IT Infrastructure on E-Filing Adoption

**Samer Aqel AbuAkel and Marhaiza Ibrahim ***

Tunku Puteri Intan Safinaz School of Accountancy (TISSA-UUM), Universiti Utara Malaysia (UUM), Sintok 06010, Kedah, Malaysia; sameraqelabuakel1982@gmail.com
* Correspondence: marhaiza@uum.edu.my

**Abstract:** Electronic filing (e-filing) adoption for tax income purposes is limited in developing countries as the practices of financial accounting and reporting, as well as digitalization in accounting systems, are more prevalent in developed countries. This paper investigates the determinants of e-filing usage in the context of emerging economies such as Jordan. Building on the Technology–Organization–Environment framework (TOE), the study proposes that the effects of relative advantage, top management support, and IT infrastructure as new variables on e-filing adoption and trust in the e-filing systems are positive. The study also proposes that trust in the e-filing system affects e-filing adoption and mediates the influence of relative advantage, top management support, and IT infrastructure on e-filing adoption. Data were collected from 315 respondents and analyzed via Smart PLS. Relative advantage and top management were found to affect the adoption of and trust in e-filing. In addition, trust in the e-filing system affects e-filing adoption and mediates the impact of relative advantage and top management support for e-filing. Therefore, decision-makers should develop a mechanism to increase trust and the benefits of using e-filing for income tax purposes.

**Keywords:** e-filing adoption; diffusion of Innovations (DOI) theory; technology–organization–environment (TOE) framework; Jordan; trust in e-filing system; financial accounting and reporting; digitalization in accounting systems

## 1. Introduction

The e-filing system is one of the e-government initiatives to submit tax returns and improve tax reporting through e-devices (Schaupp et al. 2010; Sijabat 2020). The electronic government program started in Jordan in 2003 (Sulehat and Taib 2016). The major reason behind this program is to deliver public services to the general public across the country, despite their locations. Therefore, the Income and Sales Tax Department (ISTD) initiated the e-government initiatives, which included electronic tax filing (ETF) at the beginning of 2005, given the problems associated with the manual filing system. The objective of this department is to provide a higher quality of services that lead to reduced time, as well as efforts for the general public and the workers (ISTD 2017).

The last decade has witnessed the emergence of a body of research associated with the uptake of e-government at the individual level; however, these studies have focused on developed countries (Lean et al. 2009; cited in Chaouali et al. 2016). There is a lack of research regarding the uptake of e-government in developing countries, where many projects have failed (Ozkan and Kanat 2011; cited in Chaouali et al. 2016). Among the different e-government applications, in this paper, we focus on e-filing, which allows its users to fill out and pay their taxes online. In line with Chaouali et al. (2016), only about 15% of all e-government initiatives have been successful in attaining their major goals. Carter and Bélanger (2005) stress the importance of exploring the antecedents of e-filing adoption in order to ensure its successful implementation. Only a few researchers have investigated e-tax filing from the perspective of developing countries (Akram et al. 2019). This research

attempts to respond to the recent calls for research on e-tax filing continuance from the perspective of developing countries (Lallmahomed et al. 2017; Rana et al. 2015). Hence, this research enhances the limited understanding of e-tax filing continuance intentions by theoretically developing and empirically testing an extended model for e-tax filing continuance from the perspective of developing counties.

In addition, one of the strategic priorities of public financial management reform in Jordan is to facilitate voluntary compliance by supporting the timely filing of tax declarations and increasing the use of electronic filing (Public Financial Management Reform 2018–2021 in the Hashemite Kingdom of Jordan). Furthermore, the importance of taxes for the Jordanian economy cannot be underestimated because tax revenues are the main source of government revenue for the Jordanian economy (Bawaneh 2017), which is a crucial item in the Jordanian public budget, contributing around 70% of the domestic revenue during the period from 2010 to 2016. In addition, the Kingdom's income tax income ratio to GDP stands at 15.5% (Jordan Times 2018). On the other hand, the corporate sector in Jordan plays a key role in generating government revenue. It contributes about 17.6% of total tax revenues (Jordan Strategy Forum 2018). Consequently, firms must redesign themselves towards excellence, using the required creativity and innovation tools (Antony and Bhattacharyya 2010; Shehadeh et al. 2016; Khandelwal et al. 2022).

On the other hand, although online payment is available, it is not widely used by taxpayers. Especially in Jordan, there is low usage of taxpayers using online payment. Although e-filing is available and convenient, ISTD makes an effort to promote such e-services. However, less than 10 percent of taxpayers generally use the electronic method (TADAT 2016). This will deprive Jordanian companies as end users of the benefits of electronic filing, which are the speed of refund claims compared to the manual system and the reduction of errors associated with the manual system (Koong et al. 2019).

This study employs the TOE framework through the adoption of three technological, organizational, and environmental factors as predictors of e-filing adoptions among large firms in Jordan, with trust used as a mediator between exogenous and endogenous variables. Trust is a significant factor in the user's acceptance of e-tax filling and the payment system (Sichone et al. 2018).

The world's research has focused on theories such as the Unified Theory of Acceptance and Use of Technology (UTAUT) and the Technology Acceptance Model (TAM) on e-filing adoption. There are limited researchers in the world that focus on the Technology–Organization–Environment (TOE) theory, especially in Jordan. Thus, the present study combined the Diffusion of Innovation (DOI) theory and the Technology–Organization–Environment (TOE) model in the research framework; integrating the Diffusion of Innovation (DOI) theory and the Technology–Organization–Environment (TOE) framework could help to explain the e-filing usage phenomena and examine the factors that influence e-filing adoption.

The following section reviews the studies on the benefits of e-filing and e-filing adoption. Additionally, Section 2 delves into the literature foundations and development of hypotheses for this research, while Section 3 delves into the development of a theoretical framework for this study. Section 4 explains the method for data analysis and the data collection procedure. Finally, Section 5 concludes this study by focusing on the statistical results and findings and the theoretical and practical implications, followed by a discussion of limitations and future research opportunities.

## 2. Literature Review and Hypothesis Development

The following section presents a review of relevant literature on the topic to help understand the issues raised and address them. The discussion will focus on the topics of the definition of e-filing, the benefits of e-filing, e-filing adoption, and hypotheses development.

## 2.1. E-Filing Definition

E-filing is the process of filling out tax documents through the internet with the help of software or by registering oneself on the income tax website (Kumar and Anees 2014).

## 2.2. Benefits of E-Filing

The Electronic Tax Filing (ETF) system, as an offshoot of e-government applications, is widely adopted worldwide. Owing to its significance, the Income and Sales Tax Department in Jordan applied the e-tax filing system as an offshoot of e-government to enhance efficient tax collection in Jordan. This system is beneficial. The e-filing systems allow citizens to pay their taxes at any time by using the Internet, as this facility is available 24 h a day, seven days a week (Qadar et al. 2016).

Besides the above-stated benefits, the e-filing systems reduce the operational cost at the time of a transaction to submit the tax returns. At the same time, this system saves money on paper. Furthermore, the advantage of convenience refers to minimized trips, no effort to stand in queues, and parking issues. Furthermore, the benefit of this system is that the public can transact at any time during the day. Therefore, e-filing facilitates citizens' submission of their tax returns even after hours. The benefits mentioned above of the e-tax system come back to the issue of enhancing efficiency and convenience (Kamarulzaman and Azmi 2010; Obert et al. 2018). Table 1 demonstrates the advantages of the e-filing system found in some previous studies.

**Table 1.** Benefits of e-filing adoption.

| Authors/Benefits | Save Time | Reducing Computation Errors | Saving Cost | Efficient | Safe | Quick | Environmentally Friendly | Convenient | Easy | Available |
|---|---|---|---|---|---|---|---|---|---|---|
| Tjen et al. (2019) | ✓ | ✓ | ✓ | ✓ | ✓ | ✓ | | | | |
| Singh et al. (2019) | ✓ | ✓ | ✓ | ✓ | | | | | | |
| Suharyono (2019) | ✓ | ✓ | ✓ | ✓ | ✓ | ✓ | ✓ | | | |
| Obert et al. (2018) | | ✓ | | | | | | ✓ | | ✓ |
| Arora and Gupta (2017) | | ✓ | ✓ | | | ✓ | | | | |
| Monica et al. (2017) | | | ✓ | | | ✓ | | | | ✓ |
| Aziz and Idris (2017) | ✓ | ✓ | ✓ | ✓ | | | | | | |
| Kumar and Anees (2014) | | | | | | | | ✓ | ✓ | ✓ |
| Hammouri and Abu-Shanab (2017) | ✓ | ✓ | ✓ | | | ✓ | | | | |
| Umenweke and Ifediora (2016) | ✓ | | ✓ | | | ✓ | | ✓ | | ✓ |
| Matharu et al. (2017) | ✓ | ✓ | ✓ | | | | | | | |
| Kumar and Anees (2014) | ✓ | | ✓ | | | | | | | |

## 2.3. E-Filing Adoption

E-filing is a form of e-government system utilized by authorities to improve tax collection efficiency (Alibraheem and Abdul Jabbar 2016; Nomlala and Oluka 2021). According to Mustapha et al. (2021), e-filing is a computerized system that facilitates taxpayers in submitting their income tax returns electronically to the Inland Revenue Service Board via the Integrated Tax Identification System. Arora (2016) describes it as an electronic-based procedure for filing income tax returns via the Internet (Mas'ud and Umar 2019). In addition, e-filing is the submission of tax returns via e-methods, utilizing apps and software

for preparing taxes that the related tax authorities initially approved for taxpayers, be they individuals or organizations. Its design varies according to the taxpayer's taxable income. Due to its various benefits for both taxpayers and tax collectors, the e-filing system has been globally adopted in numerous countries, and the large research community has examined the phenomenon (Lim et al. 2012; Lymer et al. 2012; Hastuti et al. 2014; Chaouali et al. 2016; Zaidi et al. 2017).

According to Soneka and Phiri (2019), adoption refers to the act of taking up or following a certain trend. Davis (1989) described two factors affecting the decision of individuals to accept innovations and systems: (1) the willingness to utilize innovations and systems that are perceived as advantageous, and (2) the willingness to utilize innovations and systems that are perceived as easy to use (Sijabat 2020). Unfortunately, despite the availability of online payment, taxpayers seldom resort to this method in developing countries. This is highly evident among Jordanian taxpayers, who are still hesitant to utilize the e-filing system despite its availability and convenience and the rampant promotional efforts conducted by the ISTD. Generally, only 10% of taxpayers have been using e-filing (AbuAkel and Ibrahim 2020). A large body of research has identified trust in government as a key variable in explaining why some people adopt new technologies while others resist (Alomari 2014; Gao and Waechter 2017; Warkentin et al. 2002). Prior studies of e-filing adoption and usage are summarized in Table 2.

**Table 2.** E-filing adoption.

| Study | Country | Respondents | Organization | DV | Independent Variable |
|---|---|---|---|---|---|
| Tjen et al. (2019) | Indonesia | 1095 respondents | Users of e-filing | E-tax filing adoption | Prior experience in offline government services; trust in government; trust in technology; trust in e-government websites; information quality; system quality; service quality; perceived usefulness; user satisfaction |
| Night and Bananuka (2019) | Uganda | 242 managers | Small business enterprises (SBEs) | E-tax system adoption | Attitude towards electronic tax system; adoption of the electronic tax system; tax compliance. |
| Veeramootoo et al. (2018) | Mauritius | 645 e-filing users | Citizens | E-filing usage | Information quality; perceived risks; system quality; service quality; confirmation; user satisfaction; continuance usage intention; habit |
| Santhanamery and Ramayah (2018) | Malaysia | 355 taxpayers | Taxpayers | Usage intention of e-filing | Security, correctness, response time, system support, availability |
| Prawati and Dewi (2018) | Jakarta, Indonesia | 1733 taxpayers | Corporate | Taxpayers | Performance expectations, system quality, and user satisfaction |
| Kumar and Sachan (2017) | India | 294 taxpayers | Citizens | Adoption of E-filing | Compatibility; image; result demonstrability; ease-of-use; perceived usefulness; trust of the Internet; trust of the government, social influence; service quality; risk |
| Matharu et al. (2017) | India | 250 respondents | Individuals | E-filing Adoption | Gender; education; age; occupation |

"An information-seeking and information-processing activity, where an individual is motivated to reduce uncertainty about the advantages and disadvantages of an innovation," as defined by Rogers (2003), is the innovation-decision process (p. 172). (1) Knowledge, (2) persuasion, (3) decision, (4) implementation, and (5) confirmation are the five stages that Rogers (2003) identifies as constituting the innovation-decision process. These steps typically occur in the order listed here. Figure 1 depicts this procedure.

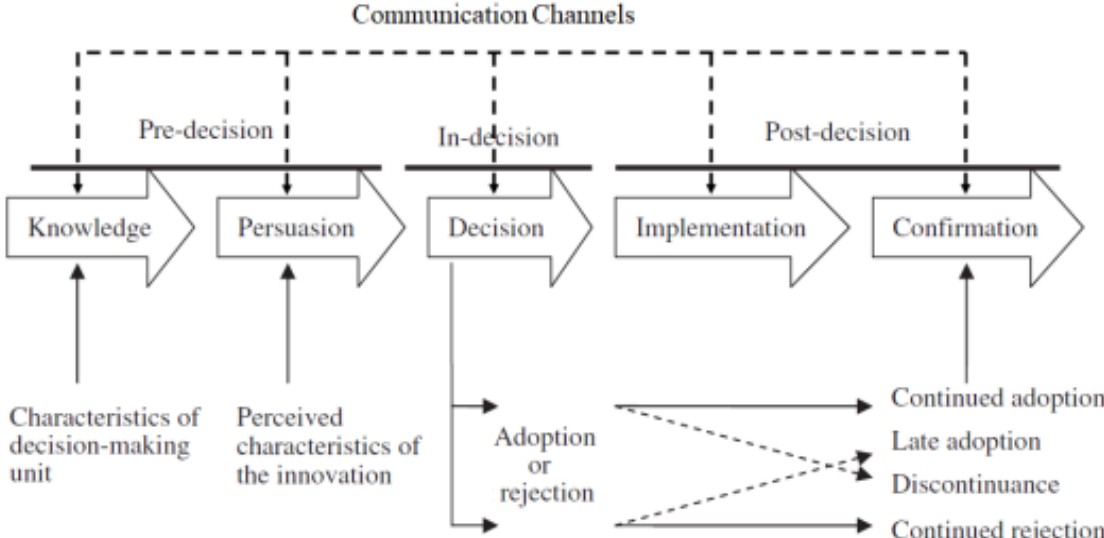

**Figure 1.** Information seeking and processing in the DOI model. Adapted from Rogers (2003).

*2.4. Hypotheses Development*

This part will deal with the literature related to TOE factors and e-tax filing adoption to illustrate hypotheses development.

### 2.4.1. Relative Advantage and Intention to Adopt E-Filing

Rogers (2003) delineated relative advantage as the perception of a certain innovation being better than its initial idea. E-Tax was created to provide a fast, simple, and non-complex online channel that delivers wide-ranging tax information and services that can be easily accessed anytime, anywhere by everyone (Ibrahim 2014; Yefni et al. 2018). Relative advantage and behavioral adoption/application were positively related (Ali et al. 2012; Daoud and Ibrahim 2019; Zhu et al. 2006).

In general, the e-filing system incorporates the aspects of preparing, filing, and paying taxes, all representing a significant benefit over the conventional method (Ambali 2009; Santhanamery and Ramayah 2018).

Other relative advantages of using e-filing include cost reduction, higher productivity, better decision-making, and time-savings (DeLone and McLean 2003; Millenia et al. 2022). On top of that, it allows pervasive access to services irrespective of location and time; it also enables the automatic estimation of the tax amount, thus reducing errors (Ullah et al. 2021). Consequently, the hypothesis below is developed:

**H1.** *Relative advantage significantly and positively affects the intention of Jordanian firms to adopt e-filing.*

When taxpayers are convinced that e-tax filing is significantly more advantageous than the conventional method, their trust in the system improves. For example, McCloskey (2006) discovered that trust is significantly related to perceived usefulness, i.e., the higher their trust, the greater their perception of the system's usefulness (Santhanamery and Ramayah 2018). Likewise, Pavlou (2003) also asserted the significance of perceived usefulness in influencing the use of e-commerce. This indicates the prevalence of a mediating role played

by perceived usefulness in the link between trust and usage intention (Santhanamery and Ramayah 2018). Consequently, this study develops the hypothesis below:

**H2.** *Relative advantage significantly and positively affects the trust of Jordanian firms in the e-filing system.*

### 2.4.2. Top Management Support and Intention to Adopt E-Filing

Top management support entails the extent to which the management can provide support to the entire team and aid in problem-solving within the given timeframe. The term indicates the degree of management support given in the aspect of new technology adoption (AbuAkel and Ibrahim 2020; Haneem et al. 2019; Premkumar and Roberts 1999). As with all new systems requiring investment in hardware and IT services, the development, launch, and maintenance of the e-filing system will require funding and resources that will need to be allocated not only for developing the software, but also for storing data, training the filers, and providing call center support services, especially during the initial launch of the system (World Bank 2018). Moreover, the top management has been deemed by scholars to be the main variable in ensuring the successful approval of new technology in an organization (AbuAkel and Ibrahim 2020; Mohtaramzadeh et al. 2018; Premkumar and Roberts 1999; Wu et al. 2003). This leads to the establishment of this hypothesis:

**H3.** *Top management support significantly and positively affects the Jordanian firms' e-filing adoption intention.*

When taxpayers, such as large firms, trust that management support will be provided in their adoption of e-filing, they will develop a more positive feeling about the system and trust its usage more. Therefore, e-government implementation success refers to top management support, such as actively participating in e-government implementation projects, thus leading to greater efficiency and effectiveness, higher trust in the government, and better support for the users (Kagoya and Mbamba 2021).

**H4.** *Top management support significantly and positively affects the Jordanian firms' trust in the e-filing system.*

### 2.4.3. IT Infrastructure and Intention to Adopt E-Filing

IT infrastructure is one of the facilitating conditions that support the use of the system. It must include technical and organizational infrastructure (Rakhmawati and Rusydi 2020; Venkatesh et al. 2003). IT infrastructure entails tangible organizational resources used in adopting new technology; IT infrastructure capability entails the organization's capability to leverage its technical and human IT resource infrastructure (AbuAkel and Ibrahim 2020; Benitez-Amado and Ray 2012).

The attributes of the ICT infrastructure are crucial in ensuring the success of e-government implementation; they entail application servers, websites, the Internet, hardware resources, and operating systems used by experts and users/citizens for creating, accessing, disseminating, and using digital information (Kagoya and Mbamba 2021; Sichone and Mbamba 2017). Thus, infrastructure, knowledge, technical facilities, and compatibility of the e-filing system with other technologies influence the usage of e-filing (Rakhmawati and Rusydi 2020).

**H5.** *IT infrastructure significantly and positively affects the Jordanian firms' intention to adopt e-filing.*

Logically, when taxpayers such as large firms believe that IT infrastructure is available to facilitate their adoption of e-filing, they will develop a more positive sentiment toward the software, thereby improving their trust in its usage. In addition, users will also develop greater confidence in the government's ability to provide a supportive environment for such transactions, considering that trust in the system would affect the users' intention to adopt the new system (Zakari et al. 2019).

**H6.** *IT infrastructure significantly and positively affects Jordanian firms' trust in the e-filing system.*

2.4.4. Trust in the E-filing System and Intention to Adopt E-Filing—Structural Effects

Trust entails the anticipation that a promise made by one party to another will be fulfilled (Rotter 1967; Zakari et al. 2019). When the targeted IT users trust the government-introduced system, they will also trust the government to prepare a proper online transaction platform (Chen et al. 2015; Zakari et al. 2019). Empirical findings indicated trust as a significant predictor of intention (Chaouali et al. 2016; Zakari et al. 2019), whereby trust towards the government was identified as a key driver of new electronic system adoption intention (Carter et al. 2011), and trust towards the system and government was a significant predictor of electronic system adoption intention (Balmi 2016; Zakari et al. 2019). According to Zaidi et al. (2017), the higher the users' trust that an e-filing system would improve business performance, the greater their intention to use it (Rakhmawati and Rusydi 2020). Consequently, this study hypothesizes:

**H7.** *Trust in the e-filing system significantly and positively affects the Jordanian firms' intention to adopt e-filing.*

Apart from the factors related to technology, organization, and environment, trust is also identified as a key factor influencing new IT innovation acceptance and diffusion among organizations (Kumar et al. 1998; Li et al. 2008, 2015; Pavlou 2002; Pavlou and Gefen 2004; Ratnasingam 2005; Schoorman et al. 2007). Additionally, trust is acknowledged to have a positive mediating effect on various contextual relations (McKnight and Chervany 1996; Li et al. 2015), facilitating a win-win cooperation strategy and hence boosting transaction efficacy. Therefore, trust is recognized as a key driver for the efficient deployment and adoption of technologies in e-government (Abu-Shanab 2017; Lallmahomed et al. 2017; Srivastava and Teo 2005; Warkentin et al. 2002). Hence, this study argues that trust has a crucial effect on e-filing adoption among Jordanian firms and thus requires further investigation. The hypotheses below are established for that purpose:

**H8.** *Trust in the e-filing system plays a significant mediating role in the link between relative advantage and the intention of Jordanian firms to adopt e-filing.*

**H9.** *Trust in the e-filing system plays a significant mediating role in the link between top management support and the intention of Jordanian firms to adopt e-filing.*

**H10.** *Trust in the e-filing system plays a significant mediating role in the link between IT infrastructure and the intention of Jordanian firms to adopt e-filing.*

**3. Theoretical Framework**

Limited studies worldwide have examined e-filing using the TOE theory. Most of the earlier studies focused on the technological standpoint instead of the business viewpoint. After careful review of the IT adoption literature, we found that the TOE framework can serve as the theoretical background unifying different factors affecting business adoption of e-tax filing (Shao et al. 2015). In addition, very few studies have examined the conceptualization of e-filing and its adoption at the organizational level (Akram et al. 2019). The majority had mainly looked at technological adoption at the individual level. In these investigations, two models have been prominently used, namely the diffusion on innovation (DOI) model, also known as the innovation diffusion theory (IDT), as well as the Technology–Organization–Environment (TOE) framework. The DOI by Rogers is widely regarded as the basis for studies on individual and organizational-level technology adoption. Three determinants have been identified to affect innovation and technology adoption decisions: innovation features, organizational features, and individual features. The DOI focuses on five innovation features that critically drive innovation diffusion, i.e., relative advantage, compatibility, complexity, trialability, and observability (Rogers 2010; Sastararuji et al. 2021). Several studies singled out complexity, relative advantage, and

compatibility as the important characteristics related to technology usage (Daoud and Ibrahim 2018; Tornatzky and Klein 1982; Wu et al. 2007). This current study is theoretically based on the Technology–Organization–Environment (TOE) framework developed by Tornatzky et al. (1990). It is a widely used model for examining the organizational adoption of novel technologies. It identifies three factors influencing an organization's decision to adopt new technologies, namely, technological, organizational, and environmental drivers. Technological drivers include all existing and new technologies pertinent to the organization. These significantly affect the organizational decision to adopt a certain innovation by shedding light on the benefits attainable by adopting the new technology. Next, the organizational drivers entail the internal organizational determinants influencing the adoption and implementation of new technology. Such factors include the organizational scope, size, structure, financial support, managerial beliefs, and top management support. Finally, the environmental drivers entail factors surrounding the organization, such as the industry, technological support infrastructure, and governmental regulations (Tornatzky et al. 1990).

Hypothesis testing was carried out to validate the theoretical framework shown in Figure 1, ultimately answering the study objective of determining the structural influences of trust on the intention of Jordanian firms to adopt e-filing. Hence, there is a crucial need to examine individual perceptions, specifically trust towards e-filing, based on past experiences utilizing the e-payment system. Therefore, there is a justifiable basis for suggesting the mediating effect of trust, leading to the proposal of the model presented in Figure 2.

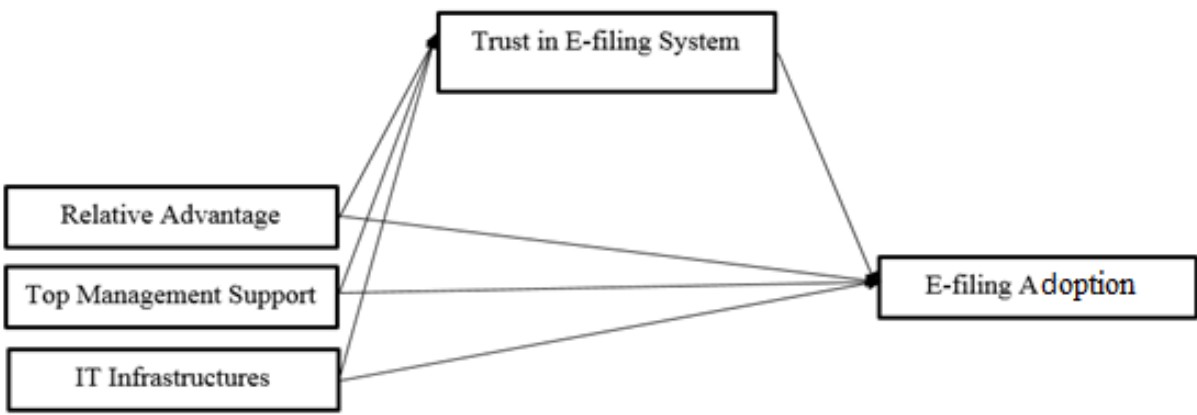

**Figure 2.** Structural mediating effects of trust in e-filing adoption.

The TOE framework is the starting point for explaining the precursors of e-government system adoption, as employed by a majority of past researchers (Hassan et al. 2017; Soufiane and Ibrahim 2021). The integration between the DOI theory and the TOE framework is, hence, projected to result in a viable theoretical framework for explaining e-government system adoption by firms (Lippert and Govindarajulu 2006; Mohamad and Ismail 2009; Ramdani et al. 2013; Soufiane and Ibrahim 2021).

Theoretically, the TOE framework and DOI theory are used in this study to identify the determinants influencing e-filing adoption by Jordanian organizations. The TOE framework (Depietro et al. 1990; Tornatzky et al. 1990) and the DOI theory (Rogers 1962) are integrated to further explore the numerous key determinants driving e-filing adoption amongst business organizations in the Jordanian context. Table 3 shows previous studies that have used DOI theory in combination with the TOE framework.

**Table 3.** DOI theory with the TOE framework.

| Study | Country | Respondents | DV | DOI and TOE Factors |
|---|---|---|---|---|
| Soufiane and Ibrahim (2021) | Algeria | Conceptual framework | E-government usage | Technological context: IT infrastructure and security Organizational context: Top management support and organization culture Environmental context: Rules and regulation awareness. |
| Hussein et al. (2019) | Jordan | 200 Manufacturing SMEs | E-commerce technology | Technological context: Relative advantage and security. Organizational context: Top management support, information intensity Mediating: Perceived usefulness |
| Van Thanh et al. (2018) | Vietnam | Employees of 397 IT departments | Adoption of e-government | Technological context: availability and characteristics. Organizational context: informal and formal linking structure, communication process size, and slack Environmental context: industry characteristics and market structure, Technology support infrastructure, and government regulation |
| Batubara et al. (2018) | Netherlands | Conceptual framework | Adoption for e-government | Technological context: availability Organizational context: organizational readiness, managerial structure, and size Environmental context: structure of the industry, technology support infrastructure, and regulatory environment |
| Shafique et al. (2017) | Pakistan | 201 staff | Adoption of e-government | Technological context: Perceived benefits, IT infrastructure, and complexity Organizational context: Organization size, top management commitment and innovativeness, and resource commitment Environmental context: External pressure, regulatory environment, and work overload |
| Ilin et al. (2017) | Western Balkan Peninsula | 276 firms | E-business adoption | Technological context: Available characteristics Organizational context: Size, slack, structure, and communication Environmental context: Industry characteristics, technology support infrastructure, and government regulation |

## 4. Research Methodology

### 4.1. Research Design, Measures, and Sampling Technique

A research design is an approach or strategy used to conduct the study, acquire data, and analyze the variables outlined in the research problem. It is essentially an outline and plans to explore the research to answer the research questions. To achieve study objectives and to answer the questions, the current study has used cause and effect empirical design by the way of a quantitative approach. Quantitative research involves the empirical and systematic investigation of phenomena by mathematics and statistics, and the result is quantitative (numerical), as researchers might not be influenced by opinions or personal feelings. The quantitative approach processes a large amount of data and allows comparisons (Basias and Pollalis 2018; Bridgmon and Martin 2012; Black 1999). In this research, a cross-sectional survey method was used. A survey investigates the relationship between various variables in the social system, including institutions, organizations, and communities. The unit of the analysis entails the studied subject. In other words, it refers to what the researcher intends to determine the empirical study on, i.e., the major focus of the study. For this current study, the unit of analysis is firms listed under the

Companies Control Department of Jordan, and managers at the upper and middle levels are considered respondents.

The present study followed a closed-ended questionnaire with two sections: the first section covered demographic information about respondents and their firms, and the second section contained questions about the research framework used in the present study. In addition, the following provides a concise overview of both subsections: Section A consists of six questions: four questions pertain to the respondent's profile, including gender, age, level of education, and field of specialization; and the remaining two questions pertain to the organization, i.e., industry type and e-filing adoption. Section B contains forty-three questions measuring five constructs adapted from previously published research studies. The survey questions for e-filing adoption were adapted from Wahsh and Dhillon (2015). The sample items for e-filing adoption are as follows: "Using e-filing services is advantageous," "E-filing is easy to use," and "E-filing ensures security and privacy." Items of trust in the e-filing system were adapted from Alsaad et al. (2017). The sample items for trust in the e-filing system are as follows: "We feel comfortable in doing business on the Internet with them," "We are comfortable in relying on them to fulfil our obligations," and "They do a good job at meeting our need." Relative advantage items were adapted from Mohtaramzadeh et al. (2018) and Safari et al. (2015). The sample items for relative advantage are as follows: "E-filing use would help increase business profitability," "E-filing provides timely information for decision making purposes," and "E-filing use would help reduce costs." IT infrastructure items were also adapted from Mohtaramzadeh et al. (2018). The sample items for IT infrastructure are as follows: "Level of sufficiently in the current Internet connection speed for E-filing transactions is" and "Level of quality e-filing applications and services which are available at increasingly affordable rates is." Top management support was adapted from Premkumar and Roberts (1999). The sample items are as follows: "The owner or manager has allocated adequate resources to adoption of online tax filing system" and "Top management is aware of the benefits of online tax filing system."

A total of 650 firms are listed under the Companies Control Department of Jordan. Therefore, the study population entails 650 firms listed under the Companies Control Department of Jordan. According to Sekaran and Bougie (2016), a sample refers to a subgroup or subset of a target population. Therefore, using the approach of Krejcie and Morgan (1970), 315 firms' samples were selected using a simple random sampling technique out of the target population of 650 firms. Simple random sampling was used to remove all hints of bias. Respondents who provide the subdivision of the larger group are selected randomly because everyone in the population set has the same probability of being chosen. Simple random sampling works well when the study's goal is to find a generalized result that can be applied to the entire population (Rahman et al. 2022).

*4.2. Data Collection Procedure*

The researcher has collected the data with adapted questionnaires from the top and middle-level managers of Jordanian firms. The questionnaire is constructed with a Likert scale of seven scales, anchored to "strongly disagree" (1) and "strongly agree" (7). The questions were structured to explore the effects of relative advantage, top management support, and IT infrastructure as new variables on e-filing adoption and trust in the e-filing systems. To be precise, the data for this study were collected for three and a half months, starting in April 2022. Considering the nature of the Jordanian firms, the survey was conducted via a Google form and face-to-face collection; the questionnaire was shared by email (addresses taken from the Jordanian firms) and WhatsApp community group managers.

### 4.3. Data Analysis Techniques

Statistical Package for Social Sciences (SPSS) version 23.0 and Smart PLS were used in this study. The data were analyzed in four phases: variance testing, factor analysis, descriptive statistics, and multiple regression analysis.

## 5. Findings

This section discusses the data examination, the profile of respondents, and the data analysis using Smart PLS.

### 5.1. Data Examination

The findings of this study were derived from SPSS and Smart PLS analyses. PLS-SEM has been an extensively acknowledged modeling technique since the advent of the 21st century, as it is a nonparametric technique for testing the research model (Salleh et al. 2016). SPSS facilitated the identification of missing values, outliers, normality, and multicollinearity that may exist in the dataset. This study involved 324 respondents. The missing value analysis showed no missing values in this study. However, nine responses were found to be outliers and, hence, were removed. This brought the total number of valid responses to 315. Normality was also checked for all the variables, which were found to have skewness and kurtosis of less than absolute 1. Table 4 presents the data analysis results.

**Table 4.** Data analysis.

| Variable | Valid | Missing Value | Skewness | Standard Error of Skewness | Kurtosis | Standard Error of Kurtosis | Tolerance | VIF |
|---|---|---|---|---|---|---|---|---|
| Relative Advantage | 315 | 0 | −0.210 | 0.137 | −0.295 | 0.274 | 0.587 | 1.704 |
| IT Infrastructure | 315 | 0 | −0.263 | 0.137 | −0.389 | 0.274 | 0.654 | 1.530 |
| Top Management Support | 315 | 0 | −0.382 | 0.137 | −0.098 | 0.274 | 0.560 | 1.516 |
| Trust in the E-filing System | 315 | 0 | −0.359 | 0.137 | −0.546 | 0.274 | 0.897 | 1.115 |
| E-filing Adoption | 315 | 0 | −648 | 0.137 | −0.156 | 0.274 | | |

### 5.2. Respondents' Profiles

Table 5 shows the respondents' profiles. The sample is shown to be dominated by males and holders of a Bachelor's degree specializing in business and administration.

**Table 5.** Respondents' Profile.

| Variable | Label | Frequency | Percent |
|---|---|---|---|
| Gender | Male | 205 | 65.1 |
| | Female | 110 | 34.9 |
| Level of Education | Diploma | 48 | 15.2 |
| | Bachelor's degree | 213 | 67.6 |
| | Master's degree | 27 | 8.6 |
| | PhD | 27 | 8.6 |
| Field of Specialization | Accounting | 28 | 8.9 |
| | Business | 139 | 44.1 |
| | Administration | 94 | 29.8 |
| | Finance | 39 | 12.4 |
| | Other | 15 | 4.8 |

### 5.3. Measurement Model

The measurement model analysis determines the factor loading, reliability, and validity of the research items. Certain items, such as items 3 and 5 under trust in the e-filing system, item 3 under relative advantage, item 3 under top management support, and item 2 under IT infrastructure, were removed. The results showed that the Cronbach's alpha (CA) and composite reliability (CR) values for all the variables are over 0.70. Furthermore, the average variance extracted (AVE) has a value over 0.50, while the square root of the AVE is higher than the cross-loading (underlined numbers in Table 6 on diagonal cells), suggesting the fulfillment of the convergent and discriminant validity (underlined in Table 6).

**Table 6.** Measurement model results.

| Constructs | CA | CR | AVE | | | | | |
|---|---|---|---|---|---|---|---|---|
| IT Infrastructure | 0.89 | 0.93 | 0.82 | <u>0.90</u> | | | | |
| E-filing Adoption | 0.88 | 0.93 | 0.80 | 0.39 | <u>0.90</u> | | | |
| Relative Advantage | 0.89 | 0.92 | 0.75 | 0.57 | 0.48 | <u>0.87</u> | | |
| Top Management Support | 0.77 | 0.75 | 0.52 | 0.39 | 0.56 | 0.44 | <u>0.72</u> | |
| Trust in the E-filing System | 0.86 | 0.92 | 0.79 | 0.23 | 0.41 | 0.29 | 0.27 | <u>0.89</u> |

### 5.4. Structural Model

The structural model assessment was carried out by determining the r-square, f-square, q-square, and path coefficient, following the recommendation of Hair et al. (2017). Figure 3 presents the results.

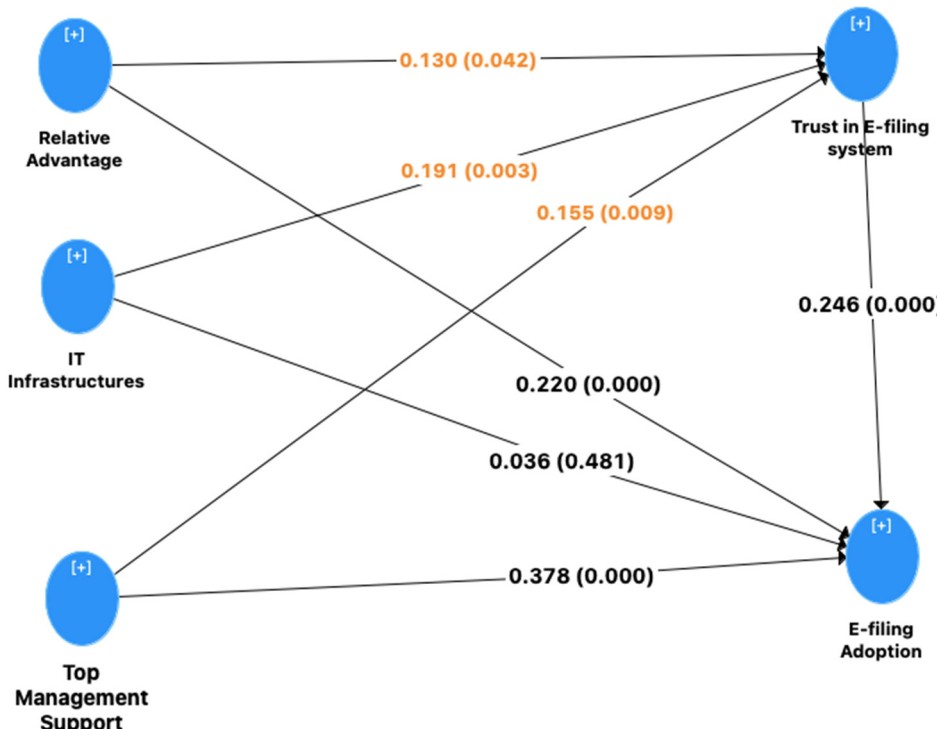

**Figure 3.** Structural model.

The model's R-square is 0.43, specifying the ability of the variables to justify 43% of e-filing adoption. In addition, the Q-square is above zero, suggesting the independent variables' ability to predict the dependent variable. Meanwhile, the F-square is higher than 0.02 except for IT infrastructure and its effect on e-filing adoption and trust in the e-filing system. Table 7 displays the findings specifically on the hypothesis (H), path,

path coefficient (B), standard deviation (Std.), t-value (T), *p*-value (P), R-square, q-square, and f-square.

**Table 7.** Hypotheses testing results.

| Path | Original Sample (O) | S.D. | T-Statistics | *p*-Value |
|---|---|---|---|---|
| IT Infrastructure → E-filing Adoption | 0.036 | 0.05 | 0.71 | 0.48 |
| IT Infrastructure → Trust in the E-filing System | 0.191 | 0.06 | 2.99 | 0.00 |
| Relative Advantage → E-filing Adoption | 0.220 | 0.06 | 3.87 | 0.00 |
| Relative Advantage → Trust in the E-filing System | 0.130 | 0.06 | 2.04 | 0.04 |
| Top Management Support → E-filing Adoption | 0.378 | 0.05 | 7.78 | 0.00 |
| Top Management Support → Trust in the E-filing System | 0.155 | 0.06 | 2.61 | 0.01 |
| Trust in the E-filing System → E-filing Adoption | 0.246 | 0.05 | 5.04 | 0.00 |
| IT Infrastructure → Trust in the E-filing System → E-filing Adoption | 0.05 | 0.02 | 2.45 | 0.01 |
| Top Management Support → Trust in the E-filing System → E-filing Adoption | 0.04 | 0.02 | 2.27 | 0.02 |
| Relative Advantage → Trust in the E-filing System → E-filing Adoption | 0.03 | 0.02 | 1.86 | 0.06 |

The first hypothesis is accepted. The effect of relative advantage on e-filing adoption is positive and significant at B = 0.21 with a *p*-value less than 0.05. Similarly, H2 is supported, and the effect of relative advantage on trust in the e-filing system is positive and significant (B = 0.17, $p < 0.05$). Additionally, supported are H3 and H4, which are related to the effect of top management support on e-filing adoption (B = 0.38, $p < 0.05$) and on trust in the e-filing system (B = 0.17, $p < 0.05$). Meanwhile, the influence of IT infrastructure on e-filing adoption is not supported. Thus, H5 is rejected.

The effect of trust on e-filing adoption is supported because B = 0.09 and the *p*-value is less than 0.05. Thus, H7 is supported. Furthermore, for H8 and H9, trust in the e-filing system mediated the effect of relative advantage and top management support on e-filing adoption. Thus, H8 and H9 are supported. Meanwhile, H10, which is related to the mediating effect of trust in the e-filing system in the link between IT infrastructure and e-filing adoption, is rejected.

### 5.5. Discussion of Findings

The current study aimed to examine the technological, organizational, and environmental factors affecting e-filing adoption among Jordanian firms. The data analysis outcome revealed that relative advantage positively influenced e-filing adoption. Hence, this hypothesis was supported, suggesting that an increase in relative advantage perception will result in e-filing adoption. Furthermore, the result was consistent with similar studies outcomes on IT adoption (Al-ghushami et al. 2018; Amini and Bakri 2015; Bhuiyan et al. 2019; Hamad et al. 2015; Ifinedo 2011).

In addition, the current study intended to determine the impact of top management support on e-filing adoption. The data analysis outcome revealed that top management support positively influenced e-filing adoption. Hence, this hypothesis was supported, suggesting that an increase in top management support will result in e-filing adoption. Furthermore, the study findings corresponded to similar study outcomes on IT adoption (Alsaad et al. 2019; Amini and Bakri 2015; Chan and Chong 2013; Ifinedo 2011; Mohtaramzadeh 2016).

On the other hand, the influence of IT infrastructure on e-filing adoption and trust in the e-filing system is not supported. The study findings corresponded to similar study outcomes on IT adoption (Wang and Feeney 2016; Luna-Reyes et al. 2008; Tolbert et al. 2008). Although the effect of trust on e-filing adoption was found to be supported, trust in the e-filing system mediated the effect of relative advantage and top management support on e-filing adoption. Meanwhile, the mediating effect of trust in the e-filing system in the link between IT infrastructure and e-filing adoption is rejected.

More taxpayers using the government's e-filing system means better government services for the public, and this research will help. However, since e-government initiatives are still relatively new in Jordan, most people have a limited understanding of what they are, how they work, and the advantages and disadvantages of conducting government business electronically.

*5.6. Contributions, Implications, and Limitations*

E-filing adoption among Jordanian firms is an essential requirement. However, since Jordanian firms try to enhance their existence in business, e-filing adoption by large firms has garnered moderate attention.

Despite the implementation and design of numerous programs by the Jordanian government to encourage the adoption of the electronic filing system, e-filing adoption continues to be a problem. Various aspects of e-filing could be addressed to ensure its adoption by providing suitable infrastructure.

It can be said that there are many contributions made by previous studies regarding e-filing at the individual level. In fact, previous studies discussed the impact of the e-filing system on the performance of individuals and the government, and mixed results were reached. Therefore, this study contributed to examining the role of trust as a mediator between the e-filing system and Jordanian companies. Moreover, the previous studies focused on the theories of TAM and UTAUT in the studies of the e-filing system, while the current study focused on the TOE theory. Therefore, the current study suggests the need to pay attention to top management support and relative advantage to activate the e-filing system.

In addition, the findings provide the Jordanian government with a deep understanding of how to develop requirements for the adoption of the e-filing system. Moreover, the results recommended that the Jordanian government design policies and programs based on technological, organizational, and environmental factors. Moreover, the study recommends that decision-makers in the Jordanian government must provide trust requirements because of their role in strengthening the relationship between organizational, technological, and environmental factors and adopting the electronic filing system. By considering the previous studies conducted around the world, particularly in developing countries with limited research on e-filing adoption with TOE theory, this study provides empirical support for the technological, organizational, and environmental factors affecting e-filing adoption. On the other hand, this study, like other research, has its limitations. Despite the current study's important practical and theoretical contributions, some limitations must be acknowledged, which should be considered in future research. The current study did not cover all factors that influence e-filing adoption, such as trust in government institutions. Future researchers might consider trust in government institutions as the driving force behind e-filing adoption and test this phenomenon in their studies. Moreover, data collection was delayed by the respondents because the nature of tax issues is sensitive. This study aimed to investigate whether there is a direct correlation between the independent and dependent variables; future studies should expand on this by examining the role that moderating constructs play in the relationship between the various independent and dependent variables. Additional research using this study's model could also investigate the trust in e-filing adoption in other developing and Arabic countries.

## 6. Conclusions

The study expanded the Technology–Organization–Environment (TOE) Theory of Adoption and Use of Technology in explaining the structural impacts of trust in the e-filing system on e-filing adoption intention amongst Jordanian firms. Through the application of structural equation modeling, relative advantage and top management support were found to have significant effects on trust in and adoption of e-filing. On the other hand, the effect of IT infrastructure on e-filing adoption and trust in the e-filing system is not supported. Therefore, trust in e-filing also influences its adoption.

**Author Contributions:** The research was undertaken independently. The authors read and approved the final manuscript. All authors have equally contributed to all phases and parts of the manuscript. All authors have read and agreed to the published version of the manuscript.

**Funding:** The authors received no financial support for the research, authorship and/or publication of this article.

**Institutional Review Board Statement:** Not applicable.

**Informed Consent Statement:** Not applicable.

**Data Availability Statement:** Available upon request.

**Conflicts of Interest:** The authors declare that the research was conducted in the absence of any commercial or financial relationships that could be construed as a potential conflict of interest.

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
