# Peer review of "The Effect of Relative Advantage, Top Management Support and IT Infrastructure on E-Filing Adoption"

_jrfm, doi:10.3390/jrfm16060295_

Round 1

Reviewer 1 Report (Previous Reviewer 2)

Revised paper may be accepted for publication by editors. 

Author Response

Thank you very much for all of your valuable suggestions.

Reviewer 2 Report (New Reviewer)

This paper expands the Technology-Organization-Environment (TOE) Theory of Adoption and uses partial least squares structural equation model to study the determinants of e-filing usage in the context of emerging economies.This study is of great significance to the research on the factors of whether to use e-filing in developing countries, and the work is solid, with the following two suggestions:

1.The expression of the article needs to be unified and standardized, such as ‘p-vale’ on page 10.

2.As for the results, some connections and applications about reality could be added. The results of the model are fully displayed in the paper, but the results do not fully explain the reality.

The quality of English Language is good.

Author Response

Thank you very much for all of your valuable suggestions. Both suggestions have been accomodated.

Reviewer 3 Report (New Reviewer)

The manuscript is quite well written, however careful reading is necessary to correct typos and also some not so clear sentences.

Author Response

Thank you very much for all of your valuable suggestions. All of the issues have been addressed in the updated manuscript.

Reviewer 4 Report (New Reviewer)

This study estimates the correlation of different technological, organizational, and environmental factors on the decision to adopt e-filing by Jordanian firms.  

The authors delineate a number of empirical hypotheses to test in a transparent way. The arguments they offer are convincing for the existence of a robust relationship between the explanatory variables and outcomes in H1-H6. However, the authors' argument for H7-H10 is weaker because of the absence to control for confounding variation in the decision to adopt e-file due to generalized trust. Therefore, further evidence of a distinct relationship between trust in e-filing adoption and the decision to use e-filing by Jordanian firms for publication.

Major Comment:

The authors provide a series of references documenting the positive association between trust in government and the adoption of new technologies, the adoption of e-commerce and, most importantly, e-filing itself (Carter, Schaupp & McBride, 2011; Balmi, 2016; Zakari, Aziz & Ali, 2019). Therefore, it is important to ensure that trust in government is not  entirely driving the decision to adopt e-filing. Trust in government and, in particular, trust in "output-side" government organizations such as trust in civil services is among the strongest predictors of voluntary tax compliance or tax morale (see Koumpias, Leonardo and Martinez-Vazquez, 2021, "Trust in Government Institutions and Tax Morale", FinanzArchiv (FA) Volume 77 (2021) / Issue 2, pp. 117-140). The authors should account of trust in government institutions such as civil services in addition to their key independent variable "trust in e-filing system". If and only if the positive results persist, then the correlation reported has merit in itself. Otherwise, omitted variable bias may be responsible for the reported effect of "Trust in E-filing System" throughout estimates for H7-H10.

Minor comment: I recommend revising the paper to drop the use of the terms "effect/impact" that, in the absence of causal empirical design, are not be warranted.

Author Response

Thank you very much for all of your valuable suggestions. All of the comments have been addressed in the updated manuscript.

Reviewer 5 Report (New Reviewer)

The author(s) improved the literature review and hypothesis development by connecting prior research to the goals of this paper. This paper makes a significant contribution to the literature and has practical applications. Therefore, I believe that it is appropriate for publication." Some additional suggestions to further enhance the clarity and readability of your statement include: Specify the specific field or area of study that the paper pertains to, as this can help provide context for the significance of its contribution. Consider elaborating on the specific ways in which the paper's findings or conclusions are relevant and applicable to real-world situations. Use more precise language to describe the paper's impact, such as using specific metrics or examples of how it has been received by other experts in the field.

The editing has been done by professional proofreaders hired by the author(s)

Author Response

Thank you very much for all of your valuable suggestions. Suggestions have been accommodated in the updated manuscript.

Reviewer 6 Report (New Reviewer)

This study examines the factors that lead to adopting e-filing in Jordan. The authors use the technology organization environment (TOE) framework and the Diffusion of Innovations (DOI) theory as a basis for developing their hypotheses. The study is informative and timely, given the growing interest in digitization globally. The comments provided here are in the spirit of improving the paper.

One of the motivations for the study described in the paper’s opening is that e-filing has experienced lower adoption in developing countries. It would be useful for the authors to provide some statistics in the introduction about the lags in adoption in developed versus developing countries, and if there are any prior studies that have examined these differences (drivers and consequences). This will make the study’s motivation clearer to the reader.

 Although the authors do this later, it would be useful in the introduction to provide the reader with the descriptions of the frameworks they discuss (TOE, UTAUT, TAM, DOI) and make it clearer to the reader the strengths of the selected frameworks relevant to this study. On page 5, the diagram from the Rogers paper helps clarify the stages of innovation adoption and that the authors are focusing on the pre-decision phases. Discussing this in the introduction would help the reader appreciate the decision to select TOE/DOI over the other frameworks and the unique contribution of this study.

The authors could be clear when they provide statements about whether such statements apply globally, or to certain countries. For example, in Line 116, the authors mention that taxpayers seldom resort to e-filing. Throughout the paper, the authors could support such statistics with citations and specifications of whether they refer to developed, developing, or global taxpayers.

The section on the theoretical framework discusses that there are limited studies that have examined TOE theory. However, no citations are provided along with the TOE discussions in this section. It would be helpful if the authors included these to help the readers refer to the other studies and see the contribution of this study clearer. The referenced citations in this section mainly belong to DOI theory.

The authors used a sampling approach to select the companies in the study. The authors could clarify why this approach was selected. For example, was it laborious to contact the full list of companies? Also, did all the companies that the authors contacted complete the survey? That would be a high response rate. Lastly, what procedures were undertaken afterward to ensure that the sample sufficiently represents the full population?

On line 422, the authors indicate that H6 is rejected. However, table 7 reflects the opposite.

The findings discussion seems brief, given the paper has 10 hypotheses and related findings.

Minor comment: The authors could consider proofreading the paper. For example, e-filing is misspelled severally (e.g., line 81) and line 82 should read 24 hours a day.

The authors could consider proofreading the paper. For example, e-filing is misspelled severally (e.g., line 81).

Author Response

Thank you very much for all of your valuable suggestions. All of the comments have been addressed in the updated manuscript.

This manuscript is a resubmission of an earlier submission. The following is a list of the peer review reports and author responses from that submission.

Round 1

Reviewer 1 Report

Investigating the determinant of e-filling adoption for tax income is Jordan is crucial because of the recent development in the Jordan economy. Particularly, when the manual approach to filling have been removed due to improve in technology. While the e-filling tax can help improve tax return, it can help improve government operation and reduce costs. I believe the study will enjoy a large audience if the authors can improve the article. 

(a)   The authors need to improve the introduction by defining the problem and highlighting their contribution at the introduction level. 

(b)  A separate section is required for a comprehensive discussion of the literature review. 

(c)   The theoretical framework needs to be changed to Technology Acceptance Model, which shows that role of technological innovation in production and software management for tax (see Yi-Shun Wang, 2003). 

(d)  What is the motivation behind the research method? 

(e)   A comprehensive discussion of finding should be included. 

(f)    The conclusion must be expanded. The authors need to include policy implication and limitation of the study.

Author Response

I appreciate the reviewers’ comments, which helped me to revise the manuscript. Most of the comments are rational, so I paid all my attention to their advices and suggestions, and the manuscript has been revised.

Reviewer 2 Report

Dear Authors, Paper needs to be more descriptive. Data analysis and its relevance need rechecking. Authors need to verify data and calculations to avoid errors if any.   Do you think that study done in Jordan will be of interest to international readers across the world? The introduction and Literature review should include references from different countries and places. You should mention the use of your study for international readers. Suggest future directions.   Literature review is not very strong and is too specific and it may ignore any new and valuable contributions in the field. It is advisable to read and refer several articles from journals like Journal of Risk and Financial Management and International Journal of Managerial and Financial Accounting, IJMFA. I ask you to refer:   https://doi.org/10.1108/FS-02-2021-0043.   Your study revolves around data analysis and qualitative analysis seems to be ignored. You must verify that the paper follows the journal format. You must eliminate Grammatical errors and improve language. Do thorough proofreading before resubmission.   Revise paper and resubmit. Best wishes!

Author Response

(The authors gave the same response as above.)

Round 2

Reviewer 1 Report

Dear Authors, 

Kindly prepared a note on our comment and how you have responded in the revised version. 

Regards, 

Reviewer

Author Response

  1. The authors need to improve the introduction by defining the problem and highlighting their contribution at the introduction level.

Response: I thank the referee for the valuable comments, and suggestions regarding introduction.  Please inform you that the whole introduction has been intensively revised according to the referee’s comments and suggestions. introduction is already improved and defining the problem and highlighting of contribution in the introduction. This is clearly shown in the third, fourth and fifth paragraphs in the introduction.

  1. A separate section is required for a comprehensive discussion of the literature review.

Response: The whole literature review has been intensively revised according to the referee’s comments and suggestions. (Benefits of e-filing, E-filing Adoption and Hypotheses Development) are already added and mentioned in the literature review.

  1. The theoretical framework needs to be changed to Technology Acceptance Model, which shows that role of technological innovation in production and software management for tax (see Yi-Shun Wang, 2003).

Response: The justification for using a (TOE) theory has been added, and this is one of the contributions of my study, because the study was conducted at the organizational level. This is clearly shown in the introduction

  1. What is the motivation behind the research method.

Response: The motivation is already added and mentioned in the research methodology.

  1. A comprehensive discussion of finding should be included.

Response: A comprehensive discussion of findings are already added and mentioned in the findings.

  1. The conclusion must be expanded. The authors need to include policy implication and limitation of the study.

Response: The conclusion is already expanded, by including it in a separate paragraph entitled Contributions, Implications and limitation.

Reviewer 2 Report

Authors were given opportunity to revise paper. But paper is not revised as per expectations.  

Author Response

  • Paper needs to be more descriptive. Data analysis and its relevance need rechecking.

Response: The paper is already been rechecked through data analysis.

  • Do you think that study done in Jordan will be of interest to international readers across the world?

Response: Yes of course, because researches in the world focused on theories (UTAUT and TAM) on e-filing adoption. Where, there are limited researches in the world that focused on TOE theory, especially in Jordan (organizational level).  Thus, the present study combined the DOl theory and the TOE model in the re-search framework, integrating DOl theory and TOE framework could help to explain the e-filing usage phenomena and to examine the factors that influence e-filing adoption. This is clearly shown in the introduction and contributions.

  • The introduction and Literature review should include references from different countries and places.

Response: Introduction and Literature review are already improved, by including separate table and figure entitled (Prior studies of e-filing adoption and usage are summarized) and (DOI theory with the TOE framework.).

  • Suggest future directions.

 Response: future directions have been suggested by including it in a separate paragraph entitled Contributions, Implications and limitation.

  • Literature review is not very strong and is too specific and it may ignore any new and valuable contributions in the field.

Response: Literature review has been revised according to the referee’s comments and suggestions.  Literature review is already improved, by including it in a separate paragraph entitled (Benefits of e-filing, E-filing Adoption and Hypotheses Development).

  • You must eliminate Grammatical errors and improve language.

Response: grammatical errors have been intensively removed by expert person.

Round 3

Reviewer 1 Report

Dear Authors, 

Thank you for sending the reversed version of your paper. 

I think the present version has been greatly improved. 

Best, 

Reviewer